# EDGERUNNER: AUTO-REGRESSIVE AUTO-ENCODER FOR ARTISTIC MESH GENERATION

**Jiaxiang Tang**[1]* **Zhaoshuo Li**[2] **Zekun Hao**[2] **Xian Liu**[2] **Gang Zeng**[1]
**Ming-Yu Liu**[2] **Qinsheng Zhang**[2]

[1]State Key Laboratory of General Artificial Intelligence, Peking University. [2]NVIDIA Research.

**https://research.nvidia.com/labs/dir/edgerunner/**

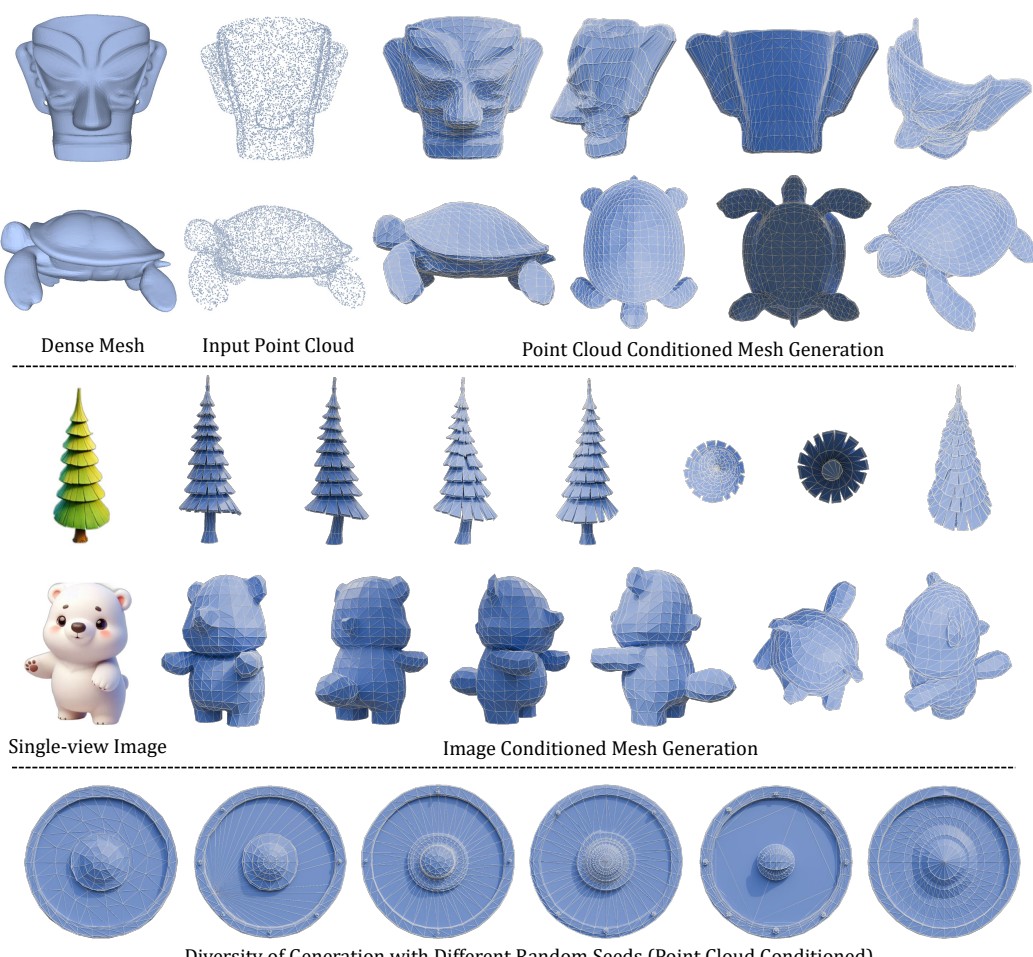

Dense Mesh    Input Point Cloud    Point Cloud Conditioned Mesh Generation

Single-view Image    Image Conditioned Mesh Generation

Diversity of Generation with Different Random Seeds (Point Cloud Conditioned)

Figure 1: **EdgeRunner** efficiently generates diverse, high-quality artistic meshes conditioned on point clouds or single-view images.

## ABSTRACT

Current auto-regressive mesh generation methods suffer from issues such as incompleteness, insufficient detail, and poor generalization. In this paper, we propose an Auto-regressive Auto-encoder (ArAE) model capable of generating high-quality 3D meshes with up to 4,000 faces at a spatial resolution of $512^3$. We introduce a novel mesh tokenization algorithm that efficiently compresses triangular meshes into 1D token sequences, significantly enhancing training efficiency. Furthermore, our model compresses variable-length triangular meshes into a fixed-length latent space, enabling training latent diffusion models for better generaliza-

---

*This work is done while interning with NVIDIA.

tion. Extensive experiments demonstrate the superior quality, diversity, and generalization capabilities of our model in both point cloud and image-conditioned mesh generation tasks.

# 1 INTRODUCTION

Automatic 3D content generation, particularly the generation of widely used polygonal meshes, holds the potential to revolutionize industries such as digital gaming, virtual reality, and filmmaking. Generative models can make 3D asset creation more accessible to non-experts by drastically reducing the time and effort involved. This democratization opens up opportunities for a wider range of individuals to contribute to and innovate within the 3D content space, fostering greater creativity and efficiency across these sectors.

Previous research on 3D generation has explored a variety of approaches. For example, optimization-based methods, such as using score distillation sampling (SDS) (Poole et al., 2022; Lin et al., 2023; Liu et al., 2023b; Tang et al., 2023a), lift 2D diffusion priors into 3D without requiring any 3D data. In contrast, large reconstruction models (LRM) (Hong et al., 2023; Wang et al., 2023b; Xu et al., 2023b; Li et al., 2023; Weng et al., 2024b) directly train feed-forward models to predict neural radiance fields (NeRF) or Gaussian Splatting from single or multi-view image inputs. Lastly, 3D-native latent diffusion models (Zhang et al., 2024c; Wu et al., 2024b; Li et al., 2024c) encode 3D assets into latent spaces and generate diverse contents by performing diffusion steps in the latent space. However, all these approaches rely on continuous 3D representations, such as NeRF or SDF grids, which lose the discrete face indices in triangular meshes during conversion. Consequently, they require post-processing, such as marching cubes (Lorensen & Cline, 1998) and re-meshing algorithms, to extract triangular meshes. These meshes differ significantly from artist-created ones, which are more concise, symmetric, and aesthetically structured. Additionally, these methods are limited to generating watertight meshes and cannot produce single-layered surfaces.

Recently, several approaches (Siddiqui et al., 2024a; Chen et al., 2024d;b; Weng et al., 2024a; Chen et al., 2024e) have attempted to tokenize meshes into 1D sequences and leverage auto-regressive models for direct mesh generation. Specifically, MeshGPT (Siddiqui et al., 2024a) proposes to empirically sort the triangular faces and apply a vector-quantization variational auto-encoder (VQ-VAE) to tokenzie the mesh. MeshXL (Chen et al., 2024b) directly flattens the vertex coordinates and does not use any compression other than vertex discretization. Since these methods directly learn from mesh vertices and faces, they can preserve the topology information and generate artistic meshes. However, these auto-regressive mesh generation approaches still face several challenges. (1) Generation of a large number of faces: due to the inefficient face tokenization algorithms, most prior methods can only generate meshes with fewer than 1,600 faces, which is insufficient for representing complex objects. (2) Generation of high-resolution surface: previous works quantize mesh vertices to a discrete grid of only $128^3$ resolution, which results in significant accuracy loss and unsmooth surfaces. (3) Model generalization: training auto-regressive models with difficult input modalities is challenging. Previous approaches often struggle to generalize beyond the training domain when conditioning on single-view images.

In this paper, we present a novel approach named **EdgeRunner** to address the aforementioned challenges. Firstly, we introduce a mesh tokenization method based on EdgeBreaker (Rossignac, 1999) that compresses sequence length by 50% and reduces long-range dependency between tokens, significantly improving the training efficiency. Secondly, we propose an Auto-regressive Auto-encoder (ArAE) that compresses variable-length triangular meshes into fixed-length latent codes. This latent space can be used to train latent diffusion models conditioned on other modalities, offering better generalization capabilities. We also enhance the training pipeline to support higher quantization resolution. These improvements enable EdgeRunner to generate diverse, high-quality artistic meshes with up to 4,000 faces and vertices discretized at a resolution of $512^3$ — resulting in sequences that are twice as long and four times higher in resolution compared to previous methods.

In summary, our contributions are as follows:

1. We introduce a novel mesh tokenization algorithm, adapted from EdgeBreaker, which supports lossless face compression, prevents flipped faces, and reduces long-range dependencies to facilitate learning.

2. We propose an Auto-regressive Auto-encoder (ArAE), comprising a lightweight encoder and an auto-regressive decoder, capable of compressing variable-length triangular meshes into fixed-length latent codes.

3. We demonstrate that the latent space of ArAE can be leveraged to train latent diffusion models for better generalization, enabling conditioning of different input modalities such as single-view images.

4. Extensive experiments show that our method generates high-quality and diverse artistic meshes from point clouds or single-view images, exhibiting improved generalization and robustness compared to previous methods.

# 2 RELATED WORK

## 2.1 OPTIMIZATION-BASED 3D GENERATION

Early 3D generation methods relied on SDS-based optimization techniques (Jain et al., 2022; Poole et al., 2022; Wang et al., 2023a; Mohammad Khalid et al., 2022; Michel et al., 2022) due to limited 3D data. Subsequent works advanced in generation quality (Lin et al., 2023; Wang et al., 2023d; Chen et al., 2023c;e; Sun et al., 2023; Qiu et al., 2024), reducing generation time (Tang et al., 2023a; Yi et al., 2023; Lorraine et al., 2023; Xu et al., 2024a), enabling 3D editing (Zhuang et al., 2023; Singer et al., 2023; Raj et al., 2023; Chen et al., 2024c), and conditioning on images (Xu et al., 2023a; Tang et al., 2023b; Melas-Kyriazi et al., 2023; Liu et al., 2023b; Qian et al., 2023; Shi et al., 2023). Other approaches first predict multi-view images, then apply reconstruction algorithms to generate the final 3D models (Long et al., 2023; Li et al., 2024b; Pang et al., 2024; Tang et al., 2024b). Recently, Unique3D (Wu et al., 2024a) introduced a method that combines high-resolution multi-view diffusion models with an efficient mesh reconstruction algorithm, achieving both high quality and fast image-to-3D generation.

## 2.2 FEED-FORWARD 3D GENERATION

With the introduction of large-scale datasets (Deitke et al., 2023b;a), more recent works propose to use feed-forward 3D models. The Large Reconstruction Model (LRM)(Hong et al., 2023) demonstrated that end-to-end training of a triplane-NeRF regression model scales effectively to large datasets and generates 3D assets within seconds. While LRM significantly accelerates generation speed, the resulting meshes often exhibit lower quality and a lack of diversity. Subsequent research has sought to improve generation quality by incorporating multi-view images as inputs(Xu et al., 2024b; Li et al., 2023; Wang et al., 2023b; He & Wang, 2023; Siddiqui et al., 2024b; Xie et al., 2024; Wang et al., 2024) and by adopting more efficient 3D representations (Zhang et al., 2024a; Li et al., 2024a; Wei et al., 2024; Zou et al., 2023; Tang et al., 2024a; Xu et al., 2024d; Zhang et al., 2024b; Chen et al., 2024a; Yi et al., 2024).

## 2.3 DIFFUSION-BASED 3D GENERATION

Analogous to 2D diffusion models for image generation, significant efforts have been made to develop 3D-native diffusion models capable of conditional 3D generation. Early approaches typically rely on uncompressed 3D representations, such as point clouds, NeRFs, tetrahedral grids, and volumes (Nichol et al., 2022; Jun & Nichol, 2023; Gupta et al., 2023; Cheng et al., 2023; Ntavelis et al., 2023; Zheng et al., 2023; Zhang et al., 2023; Liu et al., 2023c; Müller et al., 2023; Chen et al., 2023d; Cao et al., 2023; Chen et al., 2023a; Wang et al., 2023c; Yariv et al., 2023; Liu et al., 2023a; Xu et al., 2024c; Yan et al., 2024) to train diffusion models. However, these methods are often limited by small-scale datasets and struggle with generalization or producing high-quality assets. More recent approaches have focused on adapting latent diffusion models to 3D (Zhao et al., 2023; Zhang et al., 2024c; Wu et al., 2024b; Li et al., 2024c; Lan et al., 2024; Hong et al., 2024; Tang et al., 2023c; Chen et al., 2024f). These methods first train a VAE to compress 3D representations into a more compact form, which enables more efficient diffusion model training. Unlike the straightforward image representations in 2D, 3D latent diffusion models involve numerous design choices, leading to varied performance outcomes. For example, CLAY (Zhang et al., 2024c) has demonstrated that a transformer-based 3D latent diffusion model can scale to large datasets and generalize well across diverse input conditions.

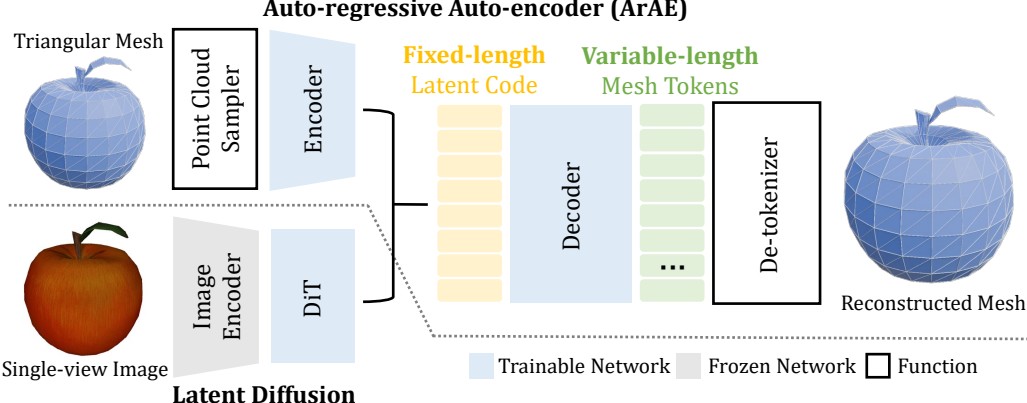

Figure 2: **Pipeline of our method**. Our ArAE model compresses variable-length mesh into fixed-length latent code, which can be further used to train latent diffusion models conditioned on other input modalities, such as single-view images.

## 2.4 AUTO-REGRESSIVE MESH GENERATION

The above works require additional post-processing steps to extract triangular meshes and fail to model the mesh topology. Recenlty, approaches using auto-regressive models to directly generate meshes have emerged. MeshGPT (Siddiqui et al., 2024a) pioneered this approach by tokenizing a mesh through face sorting and compression with a VQ-VAE, followed by using an auto-regressive transformer to predict the token sequence. This method allows for the generation of meshes with varying face counts and incorporates direct supervision from topology information, which is often overlooked in other approaches. Subsequent works (Chen et al., 2024b; Weng et al., 2024a; Chen et al., 2024d) have explored different model architectures and extended this approach to conditional generation tasks, such as point cloud generation. However, these methods are limited to meshes with fewer than 1,000 faces due to the computational cost of mesh tokenization and exhibit limited generalization capabilities. A concurrent work, MeshAnythingV2 (Chen et al., 2024e), introduces an improved mesh tokenization technique, increasing the maximum number of faces to 1,600. Our approach also falls under the category of auto-regressive mesh generation but aims to further extend the maximum face count and provide control over the target face number during inference.

## 3 EDGERUNNER

### 3.1 COMPACT MESH TOKENIZATION

Auto-regressive models process information in the form of discrete token sequences. Thus, compact tokenization is crucial as it allows information to be represented with fewer tokens accurately. For example, text tokenizers have been a central research direction for large language models (LLMs). The GPT and Llama series (Touvron et al., 2023; Brown et al., 2020) utilize the byte-pair encoding (BPE) tokenizer, which combines sub-word units into single tokens for highly compact and lossless compression.

In contrast, tokenization techniques used in prior auto-regressive mesh generation works mainly suffer from two issues. Some prior works use *lossy* VQ-VAEs (Siddiqui et al., 2024a; Chen et al., 2024d; Weng et al., 2024a), which sacrifices the mesh generation quality. Others opt for *zero-compression* by not using face tokenizers (Chen et al., 2024b), which poses training challenges due to the inefficiency.

In this paper, we introduce a tokenization scheme that allows us to represent a mesh compactly and efficiently, which is based on the well-established triangular mesh compression algorithm Edge-Breaker (Rossignac, 1999). **The key insight for mesh compression is to maximize edge sharing between adjacent triangles**. By sharing an edge with the previous triangle, the next triangle re-

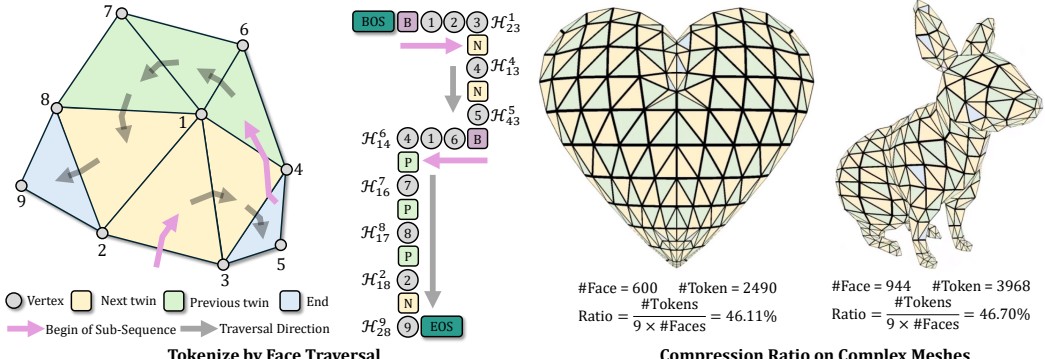

Figure 3: **Illustration of our mesh tokenizer**. Our tokenizer traverses the 3D mesh triangle-by-triangle and converts it into a 1D token sequence. Through edge sharing, we reach a compression rate of 50% (4 or 5 tokens per face on average) compared to naïve tokenization of 9 tokens per face.

quires only one additional vertex instead of three. We illustrate our mesh tokenization process with an example below, and provide more details in the appendix.

**Half-edge.** EdgeBreaker (Rossignac, 1999) uses half-edge data structures (Weiler, 1986) for triangular face traversal. An illustration is provided in Figure 4. We use $\mathcal{H}$ to denote a half-edge. For example, $\mathcal{H}_{41}^3$ is the half-edge pointing from vertex 4 to 1, with vertex 3 across the face. Starting from $\mathcal{H}_{41}^3$, we can traverse to the *next* half-edge $\mathcal{H}_{13}^4$ and the *next twin* half-edge $\mathcal{H}_{31}^2$. Reversely, the *previous* half-edge is $\mathcal{H}_{34}^1$ and the *previous twin* half-edge is $\mathcal{H}_{43}^5$. The half-edge data structure has also been used in recent learning-based mesh generation work (Shen et al., 2024).

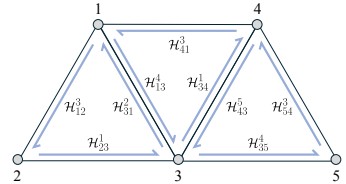

Figure 4: Half-edge representation for triangular faces.

**Vertex Tokenization.** To tokenize a mesh into a discrete sequence, vertex coordinates require discretization. Following previous works (Siddiqui et al., 2024a), we normalize the mesh to a unit cube and quantize the continuous vertex coordinates into integers according to a quantization resolution, which is 512 in this work. Each vertex is therefore represented by three integer coordinates, which are then flattened in XYZ order as tokens. With some abuse of notion, we denote the XYZ tokens as a single vertex token using ⬤.

**Face Tokenization.** We traverse through all faces following the half-edges. To illustrate the process, we use the mesh example in Figure 3. The process starts with one half-edge, where $\mathcal{H}_{23}^1$ is picked as the beginning of the current traversal. We signify the start of a traversal as B. We then append the vertex across the half-edge ① as the first vertex token. Within the same triangular face, the two vertices ② ③ are also appended following the direction of $\mathcal{H}_{23}^1$.

During traversal, we visit the *next twin* half-edge whenever possible, and only reverse the half-edge direction to the *previous twin* half-edge when we exhaust all triangles in the current traversal. Returning to the example in Figure 3, we follow $\mathcal{H}_{23}^1$ and reach $\mathcal{H}_{13}^4$. Thus, we append N to signify the *next twin* traversal direction and we only need to append ④ as ① ③ are shared. The same process is repeated for $\mathcal{H}_{43}^5$ with N ⑤ added to the current sub-sequence.

We have completed the current traversal as no adjacent faces can be found for $\mathcal{H}_{43}^5$. The sub-sequence for the current traversal is thus B ① ② ③ N ④ N ⑤.

To begin a new sub-sequence, we reversely retrieve the last-added half-edges to traverse in the opposite directions. As the last-added half-edge $\mathcal{H}_{43}^5$ doesn't have any adjacent faces, we skip it and instead consider $\mathcal{H}_{13}^4$. We go opposite to its *previous twin* half-edge $\mathcal{H}_{14}^6$. As this is a new sub-sequence, B ⑥ ① ④ are added.

We continue finding the un-visited faces in the neighborhood of $\mathcal{H}_{14}^6$ and arriving at its *previous twin* half-edge $\mathcal{H}_{16}^7$. Thus, we add P ⑦ to the current sub-mesh sequence as ⑥ ① are shared. The

process is repeated and `P` `8` `P` `2` `N` `9` are added. As all triangular faces have been visited, the face tokenization process of the mesh is complete.

**Auxiliary Tokens.** Similar to prior work in LLMs, we prepend a `BOS` at the beginning of a mesh sequence and append a `EOS` at the end of a mesh sequence.

**Detokenization.** It is straightforward to reconstruct the original mesh from a mesh token sequence. We iterate over the tokens while maintaining a state machine. Each `B` of a sub-sequence is always followed by three vertex tokens. Each `N` or `P` is followed by a single vertex token, and we retrieve two previous vertex tokens based on the traversal direction to reconstruct the triangle. Finally, we merge duplicate vertices, as they may appear multiple times from different sub-sequences, and output the reconstructed mesh.

**Advantages.** Our tokenizer benefits model training in several ways: (1) Each face requires an average of 4 to 5 tokens, achieving approximately 50% compression compared to the 9 tokens used in previous works (Chen et al., 2024b;d). This increased efficiency enables the model to generate more faces with the same number of tokens and facilitates training on datasets containing a higher number of faces. (2) Our traversal is designed to avoid long-range dependency between tokens. Each token only relies on a short context of previous tokens, which further mitigate the difficulty of learning. (3) The traversal ensures that each face's orientation remains consistent within each sub-mesh. Consequently, the generated mesh can be accurately rendered using back face culling, a feature not consistently achieved in prior methods. We will further illustrate these benefits in Section 4.

## 3.2 Auto-regressive Auto-encoder

Although our decoder is auto-regressive and generates variable-length token sequences, we observe limitations in both generation diversity and its ability to follow conditioning. In contrast, diffusion models, which have been extensively studied (Rombach et al., 2022; Zhang et al., 2024c), demonstrate promising results in addressing these challenges. To apply diffusion models, a key challenge is the requirement of fixed-length data, as mesh generation has a variable-length data structure and the number of triangle faces can vary significantly due to different complexities. Therefore, **we propose an Auto-regressive Auto-encoder (ArAE) to encode the variable-length mesh into a fixed-length latent space**, similar to the role of variational auto-encoders (VAEs) in latent diffusion models. To train our ArAE model, we choose point clouds as the encoder input for geometric information and our tokenized mesh sequences as output. Thus the ArAE model itself also works as a point cloud conditioned mesh generator.

**Architecture.** The architecture of our ArAE model is illustrated in Figure 2. ArAE consists of a lightweight encoder and an auto-regressive decoder. To extract geometric information from the mesh surface, we follow prior works to sample a point cloud and apply a transformer encoder (Chen et al., 2024d;e; Zhang et al., 2023; 2024c). Specifically, we sample $N$ random points, $\mathbf{X} \in \mathbb{R}^{N \times 3}$, from the surface of the input mesh and use a cross-attention layer to extract the latent code:

$$\mathbf{Z} = \text{CrossAtt}\big(\mathbf{Q}, \text{PosEmbed}(\mathbf{X})\big) \tag{1}$$

where $\mathbf{Q} \in \mathbb{R}^{M \times C}$ represents the trainable query embedding with a hidden dimension of $C$, PosEmbed$(\cdot)$ is a frequency embedding function for 3D points (Zhang et al., 2023), and $\mathbf{Z} \in \mathbb{R}^{M \times L}$ represents the latent code, $M < N$ and $L < C$ denote the latent size and dimension, respectively. The decoder is an auto-regressive transformer, designed to generate a variable-length mesh token sequence. For simplicity, we adopt the OPT architecture (Zhang et al., 2022), as used in prior works (Siddiqui et al., 2024a; Chen et al., 2024d;b). A learnable embedding converts discrete tokens into continuous features, and a linear head maps predicted features back to classification logits. Stacked causal self-attention layers are employed to predict the next token based on previous tokens. The latent code $\mathbf{Z}$ is prepended to the input before the BOS token, allowing the decoder to learn how to generate a mesh token sequence conditioned on the latent code.

**Face Count Condition.** Another issue is that given the same input point cloud, multiple plausible meshes with varying numbers of faces and topologies can be generated. The number of faces is particularly crucial as it directly affects the mesh's complexity (low-poly *vs.* high-poly) and the generation speed. To manage meshes with a broad range of face counts, we aim to provide some level of explicit control over the targeted number of faces. This control facilitates the estimation of

generation time and the complexity of the generated mesh during inference. We propose a simple face count conditioning method for coarse-grained control. Specifically, we append a learnable face count token after the latent code condition tokens. We bucket face count into different ranges and assign different tokens to each range. For instance, we use four distinct tokens to represent face counts in the following ranges: less than or equal to 1000, between 1000 and 2000, between 2000 and 4000, and greater than 4000. Additionally, during training, we randomly replace these tokens with a fifth unconditional token. This approach ensures that the model still learns to generate meshes without specifying a targeted face count.

**Loss Function.** The ArAE is trained using the standard cross-entropy loss on the predicted next tokens:

$$\mathcal{L}_{\text{ce}} = \text{CrossEntropy}(\hat{\mathbf{S}}[:-1], \mathbf{S}[1:]) \tag{2}$$

where $\mathbf{S}$ denotes the one-hot ground truth token sequence, and $\hat{\mathbf{S}}$ represents the predicted classification logits sequence. Additionally, to constrain the range of the latent space for easier training of subsequent diffusion models, we apply an L2 norm penalty to the latent code:

$$\mathcal{L}_{\text{reg}} = ||\mathbf{Z}||_2^2 \tag{3}$$

The final loss is a weighted combination of the cross-entropy loss and the regularization term.

### 3.3 Image-conditioned Latent Diffusion

With the fixed-length latent space provided by our ArAE architecture, it is now feasible to train mesh generation models conditioned on different inputs, akin to how 2D image generation models are trained. Among various input modalities, we showcase the capabilities of our model using single-view images, which are among the most commonly used conditions for mesh generation.

We follow the approach of previous methods (Chen et al., 2023b) and utilize a diffusion transformer (DiT) as the backbone. Specifically, we employ the image encoder from CLIP (Ilharco et al., 2021; Mohammad Khalid et al., 2022) to extract image features for conditioning. Cross-attention layers are used to integrate the image condition into the denoising features, while AdaLN layers incorporate timestep information. We use the DDPM framework (Ho et al., 2020) and mean square error (MSE) loss to train the DiT model. At each training step, we randomly sample a timestep $t$ and a Gaussian noise $\epsilon \in \mathbb{R}^{M \times L}$. The loss is calculated between the predicted noise and the random noise.

## 4 Experiments

### 4.1 Qualitative Results

**Point Cloud Conditioned Generation**. We first compare the generated meshes conditioned on point clouds in Figure 5. For the test samples, we use other image-to-3D methods (Zhang et al., 2024c; Li et al., 2024c) to generate dense meshes, ensuring that these samples have never been seen in the training dataset. Notably, MeshAnything (Chen et al., 2024d;e) uses both point coordinates and normal vectors as input to a pretrained encoder (Zhao et al., 2023). In contrast, our encoder takes only point coordinates and is trained from scratch, while still achieving better performance. Our model demonstrates greater stability on challenging test cases, producing more aesthetically pleasing meshes. Additionally, the proposed tokenization algorithm enables the generation of longer sequences, facilitating the modeling of complex shapes and improving the closure of loops for watertight surfaces.

**Image Conditioned Generation**. In Figure 6, we compare the meshes generated from single-view images. Since other auto-regressive methods cannot directly condition on images, we compare our results with a recent optimization-based approach (Wu et al., 2024a). Our end-to-end image-to-mesh generation produces artistic meshes that better capture the semantics of the input images. Additionally, our model exhibits strong generalization on challenging 2D-style or realistic-lighting images, despite being trained exclusively on images rendered from 3D meshes with simple shading.

**Face Count Control**. In practice, users may have specific requirements for the face count or level of detail in the generated mesh. Our design enables coarse-grained control over the face count by allowing users to specify the face count condition. As demonstrated in Figure 7, our model is capable of generating meshes with different ranges of face count from the same input point cloud.

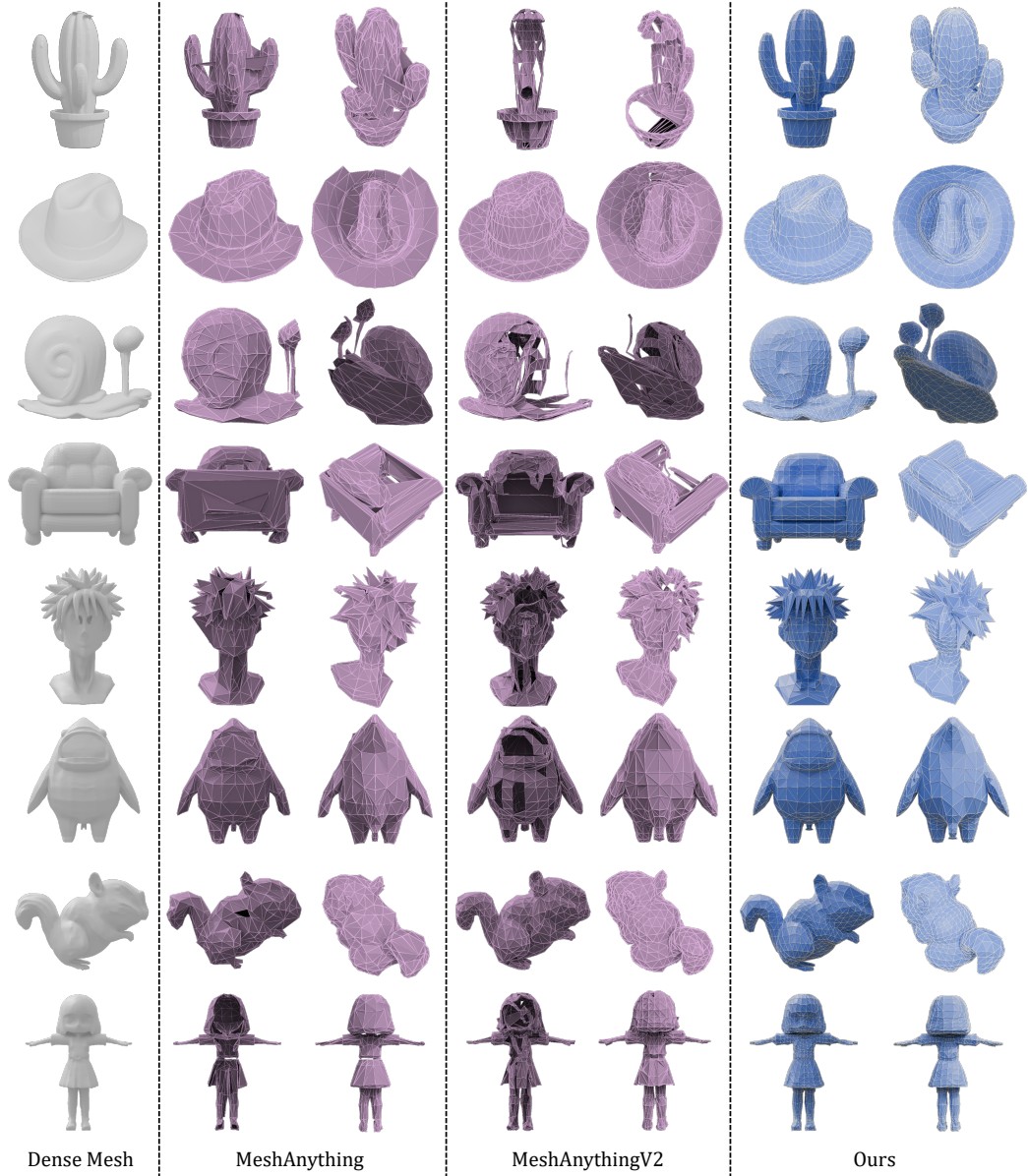

Figure 5: **Comparison on point cloud conditioned generation**. We show the reference dense mesh and generated meshes conditioned on randomly sampled point cloud.

## 4.2 QUANTITATIVE RESULTS

**User Study**. Since evaluation of the mesh aesthetic quality is challenging, we majorly rely on a user study for quantitative comparisons. For a collection of 8 test cases, we render the mesh geometry and wireframe generated from MeshAnything (Chen et al., 2024d), MeshAnythingV2 (Chen et al., 2024e), and our method. Each volunteer is shown 8 samples from mixed random methods, and asked to rate in three aspects: input consistency with the point cloud, triangle aesthetic, and overall mesh quality. We collect results from 20 volunteers with 480 valid scores. As shown in Table 1, our method is preferred across all evaluated aspects.

**Inference Speed**. Since we are using next-token prediction, the inference time for each generation is dependent on the length of the sequence. Our model is basically a large transformer, and the major compute is used for attention calculation. On a A100 GPU with flash-attention (Dao, 2024) and

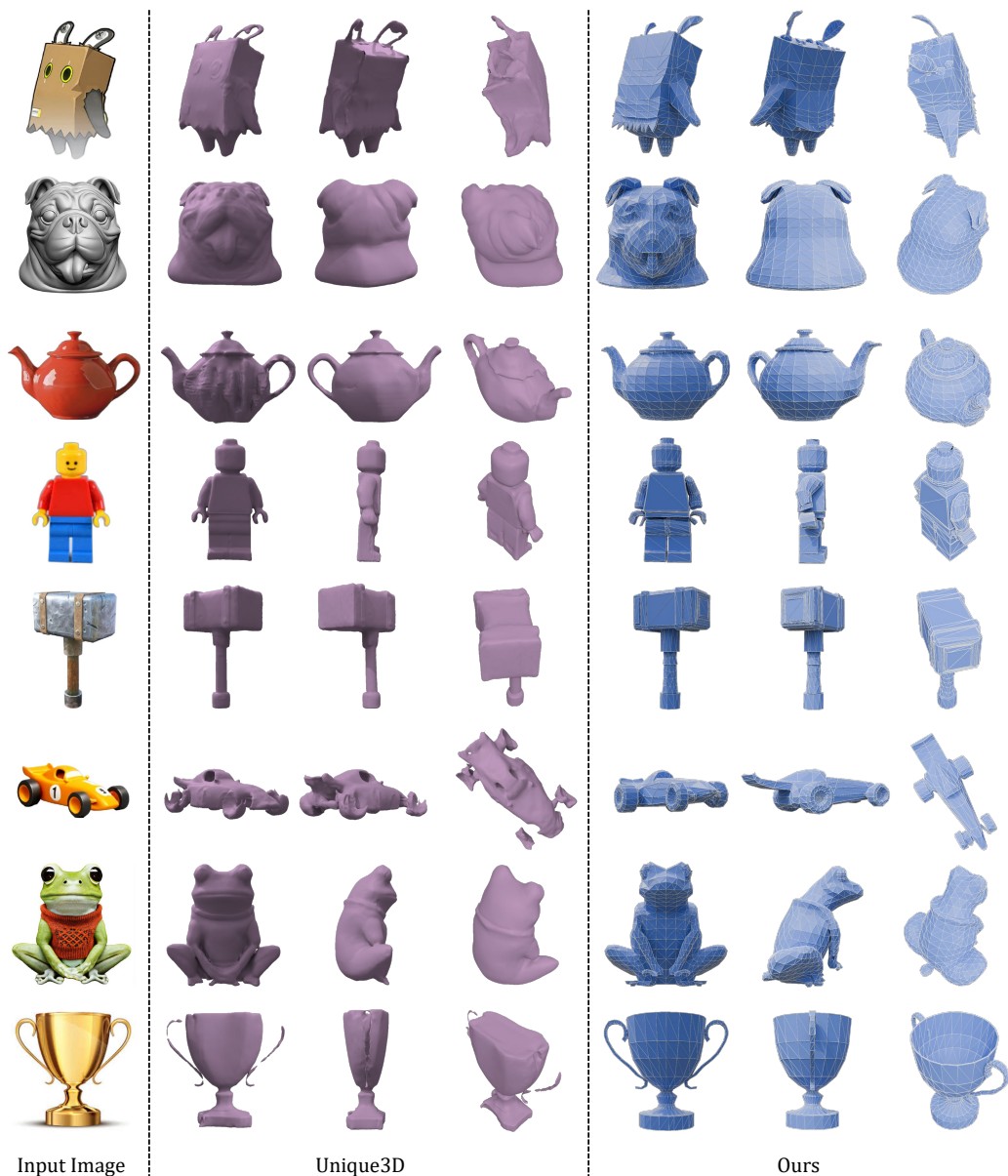

| | Input Image | Unique3D | Ours |
|---|---|---|---|

Figure 6: **Comparison on image conditioned generation**. We show the input image and generated meshes. Since Unique3D outputs dense meshes, we only visualize the surface without wireframe.

| | Input Consistency | Triangle Aesthetic | Overall Quality |
|---|---|---|---|
| MeshAnything | 2.93 | 2.43 | 2.43 |
| MeshAnythingV2 | 2.64 | 2.36 | 2.14 |
| Ours | **4.83** | **4.54** | **4.58** |

Table 1: **User Study** on point-conditioned mesh generation. The rating is of scale 1-5, the higher the better.

KV cache enabled, our model runs at about 100 tokens per second. Specifically, it takes about 45 seconds to generate a mesh with 1,000 faces, about 90 seconds for 2,000 faces, and about 3 minutes for 4,000 faces.

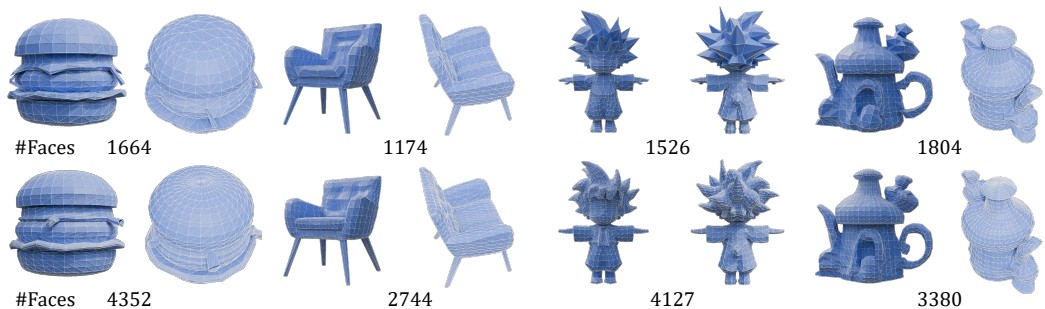

Figure 7: **Coarse-grained face count control**. Our model supports controlling the targeted number of face, which allows to generate meshes with different levels of detail.

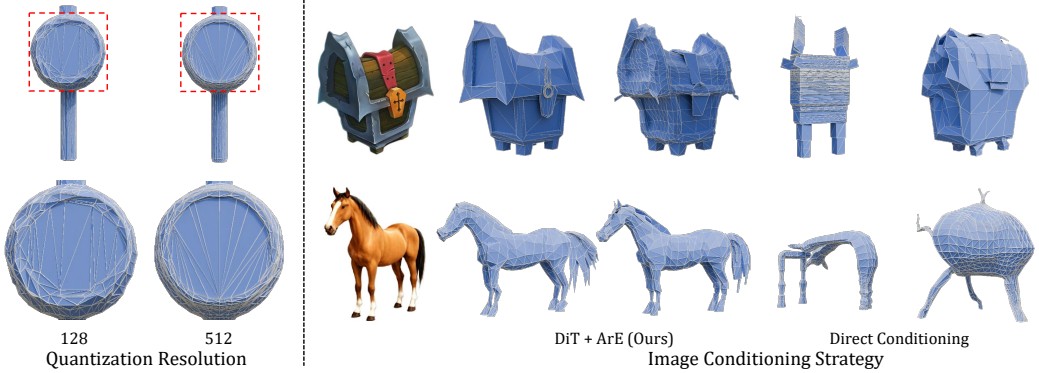

Figure 8: **Ablation Study** on different quantization resolutions and image conditioning strategies.

## 4.3 ABLATION STUDIES

**Quantization Resolution**. We initially experimented with a quantization resolution of $128^3$ as used in previous works. While this lower resolution made it easier to train the classification head, we observed noticeable quality degradation, as shown in the left part of Figure 8. To address this problem, we increased the quantization resolution to $512^3$ for our final model, which resulted in more accurate vertex positions and smoother surfaces.

**Direct Image Conditioning**. We also attempted to directly train an image-conditioned auto-regressive mesh decoder. The image features from the CLIP vision encoder were prepended as condition tokens, and the model directly predicted the mesh token sequence. However, we found that this approach was more difficult to converge and generally exhibited poorer generalization on unseen test cases, as shown in the right part of Figure 8. Consequently, we designed the ArAE model to learn a fixed-length latent space using a simpler point-cloud condition, and we trained latent diffusion models to map complex condition modalities to this latent space. This approach led to better generalization and more efficient training.

## 5 CONCLUSION

In this work, we propose a novel Auto-regressive Auto-encoder that compresses variable-length triangular meshes into fixed-length latent codes, and we further demonstrate the potential of this latent space for image-to-mesh generation. Additionally, we introduce an efficient mesh tokenization algorithm that enables scaling up the model, resulting in longer context lengths and higher resolution. Our approach highlights the potential for improving the scalability of auto-regressive models for practical 3D mesh generation, offering a promising direction for future research and applications.

ACKNOWLEDGMENTS

This work is supported by the Sichuan Science and Technology Program (2023YFSY0008), National Natural Science Foundation of China (61632003, 61375022, 61403005), Grant SCITLAB-20017 of Intelligent Terminal Key Laboratory of SiChuan Province, Beijing Advanced Innovation Center for Intelligent Robots and Systems (2018IRS11), and PEK-SenseTime Joint Laboratory of Machine Vision. We thank Yongxin Chen for introducing the EdgeBreaker paper. We thank Nicholas Sharp and Frank Shen for sharing the mesh visualization code.

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

# A APPENDIX

## A.1 IMPLEMENTATION DETAILS

### A.1.1 MESH TOKENIZER

---

**Algorithm 1:** Tokenization

---

**Data:** Discretized Vertices $\mathbf{V} = \{\mathbf{v}_i\}_N$, Triangle Faces $\mathbf{T} = \{\mathbf{t}_i\}_M$.
**Result:** Array $\mathbf{O}$ to hold the token sequence.

```
/* Write a vertex to output.                                    */
```
**def** `WriteVertex`(*Vertex* **v**)**:**
  $\mathbf{O}$.append(**v**.x, **v**.y, **v**.z);

```
/* Recursive function to compress one face.                     */
```
**def** `TokenizeFace`(*HalfEdge* **c***, bool first*)**:**
  **c.t**.$vis$ = true;
  **if** *not first***:**
    `WriteVertex`(**c.v**);
  **if** *not* **c.v**.$vis$**:**
    **c.v**.$vis$ = true;
    $\mathbf{O}$.append(N);
    `TokenizeFace`(**c.n.o**, false);
  **elif** **c.n.o.t**.$vis$ *and* **c.p.o.t**.$vis$**:**
    **return**; /* End of recursion.                           */
  **elif** **c.n.o.t**.$vis$**:**
    $\mathbf{O}$.append(P);
    `TokenizeFace`(**c.p.o**, false);
  **elif** **c.p.o.t**.$vis$**:**
    $\mathbf{O}$.append(N);
    `TokenizeFace`(**c.n.o**, false);
  **else:** /* both left and right triangles are not visited.   */
    $\mathbf{O}$.append(N);
    `TokenizeFace`(**c.n.o**, false);
    `Traverse`(**c.p.o**); /* Begin a new sub-sequence for **c.p.o.t**. */

```
/* Begin a new sub-sequence                                     */
```
**def** `Traverse`(*HalfEdge* **c**)**:**
  **if** *not* **c.t**.$vis$**:**
    $\mathbf{O}$.append(B);
    `WriteVertex`(**c.v**);
    `WriteVertex`(**c.s**);
    `WriteVertex`(**c.e**);
    **c.s**.$vis$ = 1; **c.e**.$vis$ = 1;
    `TokenizeFace`(**c**, true);

```
/* Initialization                                               */
```
**for** *Vertex* **v** *in* **V:**
  **if** **v** *is a boundary vertex***:**
    **v**.$vis$ = true;
$\mathbf{O}$.append(BOS);

```
/* Loop all unvisited faces and traverse.                       */
```
**for** *Face* **t** *in* **T:**
  **if** *not* **t**.$vis$**:**
    `Traverse`(**t**.halfedges[0]);
$\mathbf{O}$.append(EOS);

---

**Tokenization Algorithm.** To formally define the algorithm, we first describe how to use the half-edge notation to refer the mesh data in Figure 9. For a vertex **v**, its three coordinates are denoted as

$(\mathbf{v}.x, \mathbf{v}.y, \mathbf{v}.z)$. For a face $\mathbf{t}$, we store its three half-edges in an array $\mathbf{t}.halfedges$. Additionally, we use $\mathbf{v}.vis$ and $\mathbf{t}.vis$ to indicate whether this vertex or face has already been visited in the algorithm.

**Comparison.** We compare the original EdgeBreaker algorithm with our modified version and highlight the key differences. For clarity, we use N and P tokens in the main paper to represent moving to the right or left triangles (the next twin or previous twin half-edges). To remain consistent with the original algorithm, we now revert to using L and R.

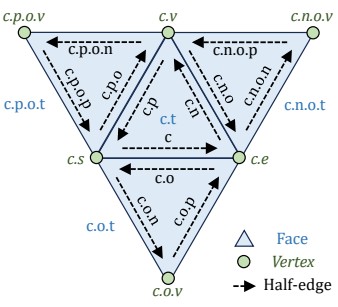

Figure 9: Mesh notation starting from half-edge $\mathbf{c}$.

(1) In the original EdgeBreaker algorithm, newly added vertices are stored in a residual form relative to the previous triangle, following the parallelogram rule (Rossignac, 1999). This method can potentially reduce the number of bits when compressed further in binary format. However, since our objective is to convert the mesh into a discrete token sequence rather than binary bits, relative vertex encoding is unnecessary. Moreover, our discretization reduces the accuracy of relative coordinates, so we simply use the absolute coordinates for each vertex.

(2) As illustrated in Figure 10, the original algorithm employs five face type tokens: C, L, E, R, and S. The special S token is used to indicate a bifurcation in the traversal tree. However, since the binary tree must be flattened into a 1D sequence, both branches dependent on the S-type triangle can introduce long-range dependencies, particularly if the first-visited branch is long, as shown in Figure 11. To avoid such dependencies, we explicitly duplicate the necessary vertices before visiting the second branch, which can be interpreted as restarting a sub-mesh sequence. While this approach slightly worsens the compression rate, it effectively eliminates long-range dependencies, which can be difficult for the transformer model to learn.

(3) During tokenization, the C token must be distinguished from the others, but it behaves similarly to the L token during de-tokenization. Since tokenization is unnecessary during inference, we replace all C tokens with L for training. Additionally, we merge the E token into B, as E always follows B after removing the S token. For simplicity, we don't consider special cases such as holes or handles, since they can still be represented at the cost of several degenerated faces, which can be easily cleaned during post-processing. As a result, our modified algorithm uses only three face types: L, R, and B.

### A.1.2 AUTO-REGRESSIVE AUTO-ENCODER

**Datasets.** For the ArAE model, we use meshes from the Objaverse and Objaverse-XL datasets (Deitke et al., 2023b;a). Given that the original datasets include many low-quality 3D assets, we filter the meshes empirically based on metadata captions or file names (Tang et al., 2024a). Our training set comprises approximately 112K meshes with fewer than 4,000 faces from both Objaverse and Objaverse-XL. For finetuning, we select only the higher-quality meshes from Objaverse, totaling around 44K. Each mesh undergoes a preprocessing pipeline where close vertices are merged, collapsed faces are removed, and the mesh is normalized to fit within a unit cube. The vertices are quantized at a resolution of $512^3$. Our tokenizer's vocabulary thus includes 512 coordinate tokens, 3 face type tokens (N, P, B) and 3 special tokens (BOS, EOS, PAD), resulting in a total vocabulary size of 518.

**Architecture.** We illustrate the detailed architecture of our network in Figure 12. The encoder of our ArAE model employs a cross-attention layer as outlined by (Zhang et al., 2023). We sample 8,192 points uniformly from the mesh surface and use a $4\times$ down-sampled embedding to query the latent code. Consequently, the latent code has a shape of $\mathbf{Z} \in \mathbb{R}^{2048 \times 64}$. The decoder transformer is comprised of 24 self-attention layers, each with a hidden dimension of 1,536 and 16 attention heads. Overall, our ArAE model contains approximately 0.7 billion trainable parameters.

**Training and Inference.** We train the ArAE model on 64 A100 (80GB) GPUs for approximately one week. The batch size is 4 per GPU, leading to an effective batch size of 256. We use the AdamW optimizer (Loshchilov & Hutter, 2017) with a cosine-decayed learning rate that ranges from $5 \times 10^{-5}$ to $5 \times 10^{-6}$, a weight decay of 0.1, and betas of $(0.9, 0.95)$. Gradient clipping is

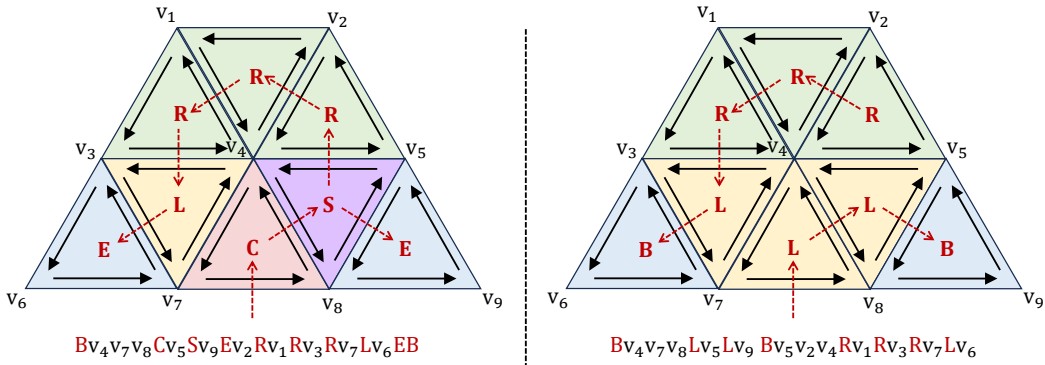

Figure 10: Comparison of the original EdgeBreaker traversal order with our modified version.

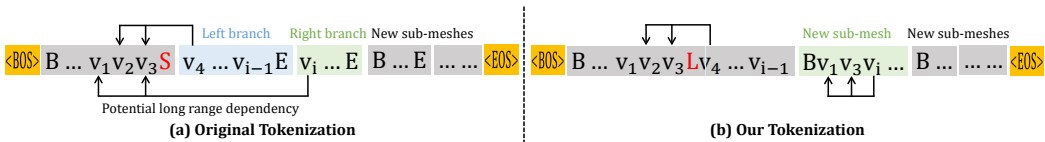

Figure 11: Our tokenization reduces potential long range dependency in the token sequence.

applied with a maximum norm of 1.0. Given that each point cloud input can correspond to multiple plausible meshes but only one ground truth mesh is available, we apply robust data augmentation techniques to address data insufficiency during training: (1) The input mesh is randomly scaled by a factor between $[0.75, 0.95]$. (2) The mesh is randomly rotated along the vertical axis by up to 30 degrees. (3) Random quadric edge collapse decimation (Garland & Heckbert, 1997) is applied to perturb the number of faces. There is a 50% chance at each iteration that the mesh will be decimated to no more than one-quarter of its original number of faces. During inference, we employ a multinomial sampling strategy with a top-k of 10. To ensure that the generated sequence is valid for decoding by the tokenizer, we apply the following rules to post-process the logits of the next token based on the prefix token sequence: (1) The token following BOS should be B. (2) After B, there should be 9 coordinate tokens. (3) After N or P, there should be 3 coordinate tokens. (4) Otherwise, the next token should be one of N, P, B, or EOS.

### A.1.3 LATENT DIFFUSION

**Datasets.** For the image-conditioned diffusion model, we use meshes from the Objaverse dataset along with rendered images from the G-Objaverse dataset (Qiu et al., 2024). The final training dataset comprises approximately 75K 3D meshes, each accompanied by rendered RGB images from 38 different camera poses. We select only 21 camera poses that align with the front face of most objects in Objaverse empirically.

**Architecture.** Our DiT model consists of 24 layers of self-attention and cross-attention, as detailed in Section 3. Each layer can be expressed as follows:

$$\gamma_1, \beta_1, \alpha_1, \gamma_2, \beta_2, \alpha_2 = \text{Chunk}(\mathbf{T} + \mathbf{t}_e) \tag{4}$$

$$\mathbf{x} = \mathbf{x} + \alpha_1 \times \text{SelfAttention}(\text{LayerNorm}_1(\mathbf{x}) \times (1 + \beta_1) + \gamma_1) \tag{5}$$

$$\mathbf{x} = \mathbf{x} + \text{CrossAttention}(\mathbf{x}, \mathbf{c}) \tag{6}$$

$$\mathbf{x} = \mathbf{x} + \alpha_2 \times \text{FeedForward}(\text{LayerNorm}_2(\mathbf{x}) \times (1 + \beta_2) + \gamma_2) \tag{7}$$

where $\mathbf{T}$ is a layer-wise learnable scale shift table, $\mathbf{t}_e$ denotes the timestep features, and $\mathbf{c}$ denotes the condition features. All attention layers have a hidden dimension of 1,024 and use 16 attention heads. We utilize the pretrained CLIP-H model from OpenCLIP (Ilharco et al., 2021) as the image feature encoder. Features from the last layer of the vision transformer are employed as conditions for the cross-attention layers. The DiT model has approximately 0.5 billion trainable parameters in total.

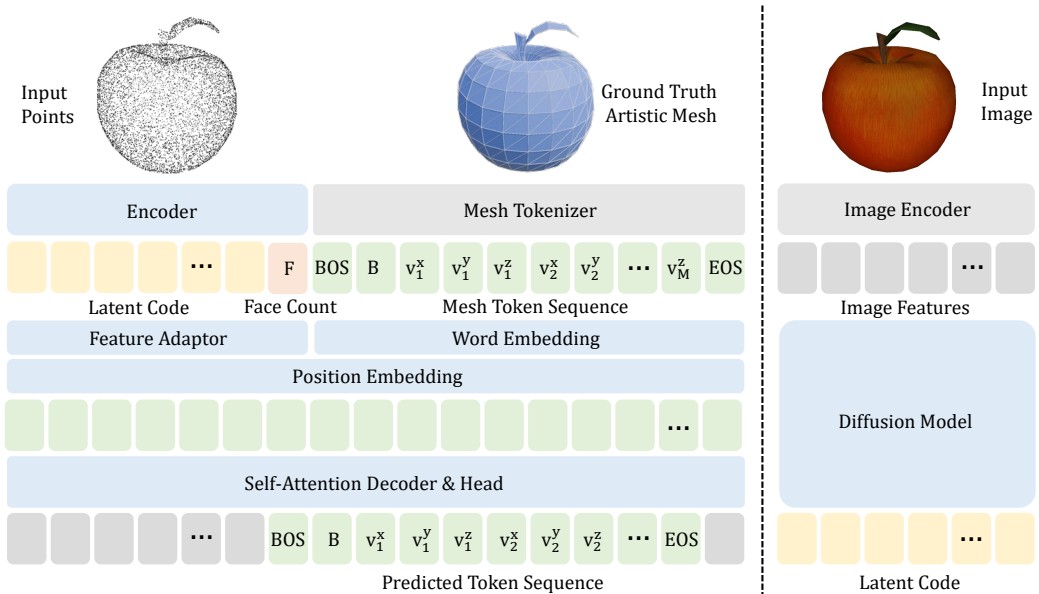

Figure 12: **Network Architecture** of our ArAE and DiT models. Our ArAE contains an encoder that takes a point cloud as input and encodes it to a fixed-length latent code, and an auto-regressive decoder that predicts variable-length mesh token sequence. The latent space can be used to train diffusion models conditioned on other more difficult conditions, such as single-view images.

|  | AI-Mesh | | Artist-Mesh | |
|---|---|---|---|---|
|  | CD↓ | HD↓ | CD↓ | HD↓ |
| MeshAnything | 0.072 | 0.170 | 0.066 | 0.195 |
| MeshAnythingV2 | 0.081 | 0.241 | 0.050 | 0.164 |
| Ours | **0.034** | **0.079** | **0.036** | **0.112** |

Table 2: **Reconstruction Quality** on point-cloud conditioned mesh generation.

**Training and Inference.** We train the DiT model on 16 A100 (40GB) GPUs for approximately one week. The lightweight ArAE encoder allows for on-the-fly calculation of the latent code during training, so we can also apply random down-sampling and rotation as data augmentation techniques for the point cloud input. The batch size is set to 32 per GPU, resulting in an effective batch size of 512. We use the same optimizer as for the ArAE model. Training employs the min-SNR strategy (Hang et al., 2023) with a weight of 5.0. The DDPM is configured with 1,000 timesteps and utilizes a scaled linear beta schedule. For classifier-free guidance during inference, 10% of the image conditions are randomly set to zero for unconditional training. During inference, we use the DDIM scheduler (Song et al., 2020) with 100 denoising steps and a Classifier-Free Guidance (CFG) scale of 7.5.

## A.2 MORE RESULTS

### A.2.1 RECONSTRUCTION QUALITY

We conducted a qualitative comparison experiment to evaluate the Chamfer Distance (CD) and Hausdorff Distance (HD) between the generated meshes and the input point clouds. Due to the lack of widely recognized evaluation datasets, we tried to construct two test datasets from different sources: (1) AI-Mesh: This dataset comprises 30 meshes generated by other image-to-3D models, extracted using the Marching Cubes algorithm. (2) Artist-Mesh: This dataset consists of 100 artistic meshes obtained from basemesh[1], which were not seen during the training process. All meshes are

---

[1]https://www.thebasemesh.com/

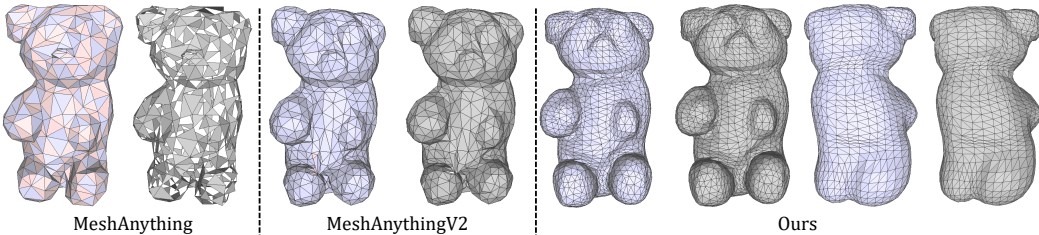

MeshAnything       MeshAnythingV2                    Ours

Figure 13: **Face orientation**. Our model ensures correct face orientation of the generated mesh.

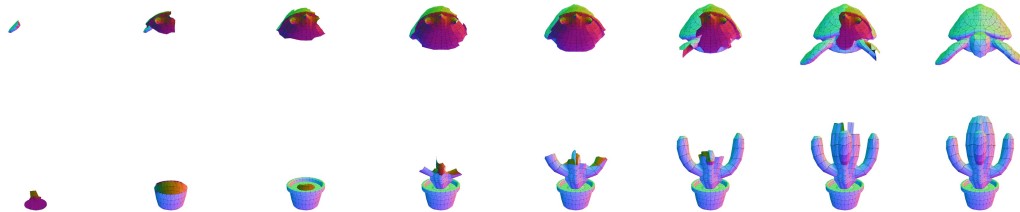

Figure 14: Visualization of the generation progress. Please visit our project page for video results.

| | Compression Ratio ↓ | Sub-sequence Count ↓ | Tokenization Speed ↑ |
|---|---|---|---|
| AMT | **46.2%** | 199.5 | 8.6 |
| Ours | 47.4% | **54.7** | **25.2** |

Table 3: **Comparison of the tokenizer algorithm.** Our tokenizer has a comparable compression ratio as AMT (Chen et al., 2024e) but uses much fewer sub-sequences per mesh, which makes model training easier. Moreover, our implementation is three times faster (mesh per second).

normalized into $[-1, 1]^3$. The summarized evaluation scores are presented in Table 2. We observed the following: (1) Our model achieves the best performance on surface metrics, even when using only point clouds as input. In contrast, the MeshAnything (Chen et al., 2024d) series further leverage surface normals. (2) MeshAnythingV2 (Chen et al., 2024e) performs significantly worse on the more challenging AI-Mesh dataset, often failing to generate complete surfaces in these cases (consistent with the results shown in Figure 5).

### A.2.2 FACE ORIENTATION

In Figure 13, we show the face orientation and rendering with back-face culling. The tokenization algorithm of our method and MeshAnythingV2 (Chen et al., 2024e) ensures that the face orientation in each mesh sequence is correct, while MeshAnything (Chen et al., 2024d) fails to do this.

### A.2.3 GENERATION PROGRESS

In Figure 14, we show the generation progress by simultaneously de-tokenizing the sequence. Our model successfully learns how to generate a valid sequence for the modified EdgeBreaker algorithm to de-tokenize.

### A.2.4 TOKENIZATION ALGORITHM

In Table 3, we compare our mesh tokenization algorithm with AMT from MeshAnythingV2 (Chen et al., 2024e). Both algorithms aim to traverse all triangular faces. While AMT can only move to a fixed side that shares the last two vertices in the sequence, our algorithm allows movement to both the left and right sides. This flexibility reduces the number of sub-sequences, making training easier and improving model robustness, as demonstrated in Figure 5. Although AMT requires only

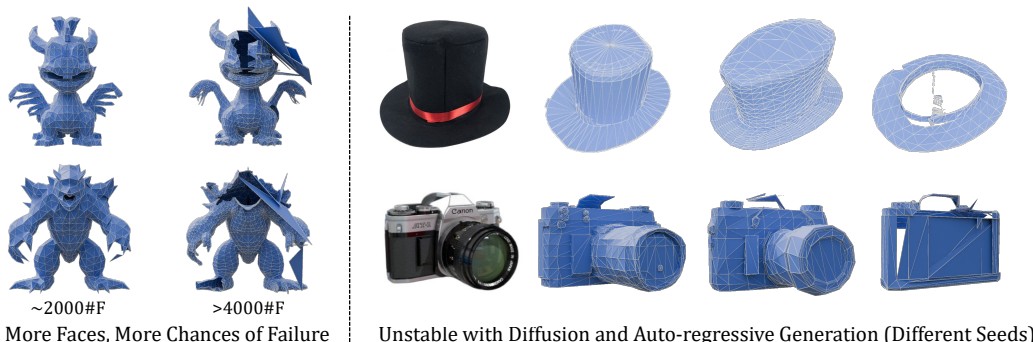

~2000#F        >4000#F
More Faces, More Chances of Failure     Unstable with Diffusion and Auto-regressive Generation (Different Seeds)

Figure 15: **Limitations**. We show some failure cases of our method to illustrate our limitations.

3 coordinate tokens for a subsequent face and our method requires 4 tokens (with an additional face type token), the amortized compression ratio remains similar.

### A.2.5 LIMITATIONS

Despite the promising results, our approach has several limitations: (1) The compression ratio of our tokenization algorithm remains insufficient. Complex meshes, such as those representing game characters, typically require a much higher number of faces, exceeding the current capacity of our model. (2) Although our model can generate meshes with up to 4,000 faces, the success rate decreases as the sequence length increases, and the generation time becomes progressively longer. (3) While the auto-regressive next-token prediction allows for diverse outputs from the same input, it also leads to potentially unsatisfactory results. Since there are infinite plausible face topologies for the same surface, precise control over the generation is difficult to achieve. We illustrate some failure cases in Figure 15.

