# OpenReview forum: "EdgeRunner: Auto-regressive Auto-encoder for Artistic Mesh Generation"
_ICLR.cc/2025/Conference — ICLR 2025 Poster_

### Official Review · Reviewer_Nd3B · 2024-10-31

**Soundness:** 4
**Presentation:** 3
**Contribution:** 4
**Rating:** 8
**Confidence:** 5

**Summary:**

This paper proposes an Auto-regressive Auto-Encoder (ArAE) model for multi-modality mesh generation supporting point clouds and single-view image inputs. Besides, an efficient mesh tokenization inspired by the EdgeBreaker algorithm is designed for more edge sharing during compression.

**Strengths:**

(i) The idea is very novel. Adapting the EdgeBreaker algorithm for mesh token compression is super interesting with the key insight "maximize edge sharing between adjacent triangles". It is a kind of Renaissance. I really like this style. This solution avoids the use of some lossy VQ-VAE models and preserves the geometry information during tokenization. Meanwhile, the customized traversal can also avoid the computation of long-range dependencies and ensure the face's orientation consistency in each sub-mesh. In addition, the core idea of using ArAE to handle variable-length mesh into fixed-length latent tokens is awesome. This can support the multi-modality inputs and address the variable-length issue in mesh generation.

(ii) The presentation is well-dressed. The writing is clear and easy to follow. The pipeline in Figure 2 is simple and clear to show the workflow of the proposed method. The visual comparison of point cloud conditioned mesh generation in Figure 5 and image conditioned mesh generation in Figure 6 are very clear to demonstrate the advantages of the proposed method over state-of-the-art algorithms.

(iii) The performance is very solid. The proposed method not only achieves more visually pleasant generation results with the highest user study score but also yields a much higher compression ratio to boost the model's efficiency. The experiments are sufficient. Especially the user study in Figures 7 and 8. It seems like more faces lead to finer-grained generated meshes. I also like the style of the visual ablation in Figure 8, studying the effect of resolution and image conditioning strategies.

(iv) The implementation details are all provided in the appendix. I believe other researchers can easily reproduce the results. I trust the authors for the reproducibility.

**Weaknesses:**

Just some minor issues:

(i) It would be better to remove the ablation study table from the supplementary to the main table as there are no other quantitative evaluation results. Although the visual comparison is good, I think the main paper should have some numerical comparison.

(ii) Maybe Figure 4 and the left part of Figure 3 should be in the same figure according to the description in Section 3.1. I think Figure 4 , instead of the right part of Figure 3, is more related to the left part of Figure 3.

(iii) For better understanding, in Line 320, the authors should add some description where to introduce the image condition input since in Line 298 - 302, the ArAE is introduced as a point clouds conditioned mesh generator.

(iv) A small typo: (low-poly *v.s.* high-poly) $\rightarrow$ (low-poly *vs.* high-poly)

**Questions:**

I am curious about the engineering tricks the authors used to save GPU memory and scale up the training datasets.

---

> ### Author Response · Authors · 2024-11-19
> **Response to Reviewer Nd3B**
>
> Thank you for your valuable time and insightful comments! We have tried to address your concerns in the updated manuscript and our rebuttal text:
>
> **Q1: Presentation.**
>
> Thank you for the suggestion! We have updated the manuscript as follows:
>
> - Moved the user study table into the main paper for more quantitative comparison.
> - Clarified the discussion on image-conditioned generation when introducing face number control to avoid confusion.
> - Corrected the typo in "vs."
>
> **Q2: Engineering tricks to scale up training.**
>
> Thank you for your interest! We employ standard engineering techniques, including flash attention, gradient checkpointing, and `bfloat16` mixed-precision training to optimize GPU memory usage. With these optimizations, each batch requires approximately 20 GB of GPU memory, accommodating up to 4,000 faces (roughly 20,000 tokens per sequence).
>
> We promise to release the code soon to facilitate better understanding and reproducibility of our method.

---

> > ### Author Response · Authors · 2024-11-25
> > **Response to Reviewer Nd3B**
> >
> > We sincerely appreciate your great efforts in reviewing this paper. Your constructive advice and valuable comments really help improve our paper. Considering the approaching deadline, please, let us know if you have follow-up concerns. We sincerely hope you can consider our reply in your assessment, and we can further address unclear explanations and remaining concerns if any.
> >
> > Once more, we are appreciated for the time and effort you've dedicated to our paper.

---

### Official Review · Reviewer_vyBt · 2024-11-02

**Soundness:** 3
**Presentation:** 3
**Contribution:** 3
**Rating:** 8
**Confidence:** 3

**Summary:**

The paper introduces a novel mesh tokenization algorithm The tokenization supports lossless face compression and reduces long-range dependencies. This helps in smooth and efficient training of an auto regressive encoder termed as (ArAE). This encoder is capable of compressing point clouds to into fixed-length latent codes representing meshes. ArAE's latent space can be leveraged to train latent diffusion models for better generalization, enabling conditioning of different input modalities such as single-view images. Hence, the method can generate meshes from point clouds or single-view images, showing generalization and multimodal capabilities .

**Strengths:**

1) The method of tokenization is based on the half edge data structure and is able to achieve 50% more compression than the existing method. This allows for smoother training and avoiding long range dependencies.

2) The paper shows generalization using single-image to mesh examples. The examples shown in the paper are of high quality.

3) The paper shows that the fixed latent space learned by the autoregressive model can be used by the diffusion training to learn many modalities including single-image to mesh capabilities.

**Weaknesses:**

1) The paper talks about the point cloud sampler but not enough information is provided on it. How is the current method sensitive to the number of points , regions of missing points, and irregularities in the point cloud positions?

2) How is the Siddique et.al's method compared with the output quality? The paper mentions the method in context of the tokenization is lossy reconstruction but does not provide the evidence of the final output quality.

3) The quality of the single-image to meshes would be more exciting to see if the poses of the input images are varied. For example, dog and frog examples in Figure 6, and compared with the comparing method.

**Questions:**

Please refer to the questions in weakness section.

---

> ### Author Response · Authors · 2024-11-19
> **Response to Reviewer vyBt**
>
> Thank you for your valuable time and insightful comments! We have tried to address your concerns in the updated manuscript and our rebuttal text:
>
> **Q1: Lack of details on the point cloud sampler.**
>
> Thank you for the reminder! We simply sample 8192 random points from the surface of the input mesh.
>
> Since the primary use case for the point-conditioned model is retopology of an existing dense mesh, we introduce random surface perturbations by adding Gaussian noises. This enhances robustness to noisy or irregular surfaces, which are commonly encountered in meshes generated by the Marching Cubes algorithm.
>
> However, we do not apply augmentations related to point count or missing regions, making the model less effective in such scenarios. Designing augmentations based on the actual data distribution could possibly enable the model to learn to predict with fewer points or handle missing regions.
>
> **Q2: Comparison with MeshGPT and VQ-VAE based tokenizer.**
>
> It is challenging to make a fair comparison with MeshGPT due to differences in training settings. For example, MeshGPT is trained on single-category data from ShapeNet (e.g., tables, chairs), while our model is trained on general objects from Objaverse. Additionally, MeshGPT quantizes vertices at a resolution of $128^3$, which results in significant information loss, while we use a finer resolution of $512^3$.
>
> Previous studies have also explored the performance of VQ-VAE. For example, MeshAnything continues to use VQ-VAE but reports that avoiding compression (retaining 3 tokens per vertex) yields better results compared to MeshGPT's default setting of 2 tokens per vertex. Both MeshXL and MeshAnythingV2 have abandoned VQ-VAE entirely, as they found compression to negatively affect performance.
>
> **Q3: Results on image-to-mesh generation with varied image poses.**
>
> Thank you for the suggestion! During training, we align the generated mesh with the pose of the input image, following practices from prior image-to-3D methods.
>
> Most test images, however, are 2D paintings collected from the web, making it challenging to alter the pose of the same object. Nonetheless, our project page includes 52 image-conditioned samples, showcasing more challenging poses, such as side or top views.

---

> > ### Author Response · Authors · 2024-11-25
> > **Response to Reviewer vyBt**
> >
> > We sincerely appreciate your great efforts in reviewing this paper. Your constructive advice and valuable comments really help improve our paper. Considering the approaching deadline, please, let us know if you have follow-up concerns. We sincerely hope you can consider our reply in your assessment, and we can further address unclear explanations and remaining concerns if any.
> >
> > Once more, we are appreciated for the time and effort you've dedicated to our paper.

---

> > > ### Comment · Reviewer_vyBt · 2024-11-26
> > >
> > > Thanks you authors. The authors have addressed my concerns. I retain my rating.

---

### Official Review · Reviewer_3xAw · 2024-11-03

**Soundness:** 4
**Presentation:** 3
**Contribution:** 3
**Rating:** 8
**Confidence:** 4

**Summary:**

The paper proposes a method for 3d mesh generation. They propose a mesh serialization copression algorithm to tokenize 3d meshes. An encoder-auto regressive decoder network operating in this representation is trained to compress this representation into a fixed length latent space. Additionally the authors propose a latent diffusion approach for generating 3d mesh representations in this latent space conditioned on images and 3d point clouds.

**Strengths:**

The proposed mesh tokenization based on maximizing edge sharing between adjacent triangles is sensible, well referenced and very useful in practice.  It supports lossless compression at 50% rate designed to reduce long range dependencies between tokens which is beneficial for learning.

The proposed pipeline solves several issues of 3d mesh generation such as the ability to generate variable length outputs through the AR decoder combined with a fixed length latent code which enables the use of diffusion models for conditional generation

Generated examples are shown to be of great quality.

Literature review is comprehensive.

**Weaknesses:**

There is no discussion about pretraining the model, loss curves showing training progression, or explaining how unconditional generation works compared to the conditional method.

**Questions:**

Is it possible to generate unconditional samples using your approach?

---

> ### Author Response · Authors · 2024-11-19
> **Response to Reviewer 3xAw**
>
> Thank you for your valuable time and insightful comments! We have tried to address your concerns in the updated manuscript and our rebuttal text:
>
> **Q1: Lack of discussion about training progress such as loss curves.**
>
> Thank you for the reminder! Our model is initially trained on approximately 112K data samples from Objaverse and Objaverse-XL. Subsequently, it is fine-tuned using 44K higher-quality samples from Objaverse. During the first training stage, the loss decreases from around 6 to 0.37, while the second stage further reduces the loss to 0.32. Notably, our data augmentation increases the training loss initially; however, the final performance proves to be more robust upon convergence.
>
> **Q2: Is it possible to perform unconditional mesh generation?**
>
> Yes, we conducted early-stage experiments with unconditional generation. Instead of prepending conditional tokens to the sequence, we trained an auto-regressive model using only mesh tokens. However, we found that unconditional generation has limited practical applications. Unlike language models, which are inherently suited for completion tasks, mesh sequences produced by EdgeBreaker are not easily prepared as the input. Consequently, we chose to focus on conditional generation from point clouds or single-view images.

---

> > ### Author Response · Authors · 2024-11-25
> > **Response to Reviewer 3xAw**
> >
> > We sincerely appreciate your great efforts in reviewing this paper. Your constructive advice and valuable comments really help improve our paper. Considering the approaching deadline, please, let us know if you have follow-up concerns. We sincerely hope you can consider our reply in your assessment, and we can further address unclear explanations and remaining concerns if any.
> >
> > Once more, we are appreciated for the time and effort you've dedicated to our paper.

---

> > > ### Comment · Reviewer_3xAw · 2024-11-26
> > > **Thank you for responding to my review**
> > >
> > > The rebuttal has adressed my queries and I choose to keep my score.

---

### Official Review · Reviewer_8Uu4 · 2024-11-03

**Soundness:** 4
**Presentation:** 3
**Contribution:** 4
**Rating:** 6
**Confidence:** 4

**Summary:**

This paper introduces EdgeRunner, an auto-regressive mesh generative model that presents a novel and efficient mesh tokenization algorithm capable of handling a higher number of mesh faces and higher resolutions. By encoding variable-length meshes into a unified latent space, the proposed method can generate meshes from point clouds or single-view images. The paper demonstrates a well-motivated and effective approach, creating a cohesive narrative.

However, key information, such as training data and experimental settings, is missing from the main text, which limits readability. Given the novelty and effectiveness of the method, I suggest accepting the paper, but I strongly recommend that the authors restructure the content to improve readability and accessibility.

**Strengths:**

The paper proposes a novel method for mesh tokenization which allows to handle mesh with more faces and in higher resolution.  It also introduces a latent space with a unified token length, enhancing the model's generalization ability. These two contributions represent the technical novelties of the paper.

The paper includes a comprehensive comparison with several state-of-the-art mesh generative models, such as MeshAnything, MeshAnythingV2, and Unique3D. This thorough evaluation makes the study well-rounded and complete.

The method is clearly described and easy to follow. The figures and renderings are thoughtfully designed, giving the paper a polished visual appeal. Additionally, the accompanying website provides extensive results, further illustrating the method's effectiveness and aiding in comprehension of its impact.

**Weaknesses:**

While I find no weaknesses in the method itself, I believe the presentation of the paper needs restructured. Important information, such as experimental settings and training datasets, currently appears only in the supplementary materials, which makes the paper difficult to follow.

Presentation:
- Line 102: The term "EdgeBreaker" appears for the first time without a reference, making this part challenging to understand.
- Line 200: It seems that the mesh tokenization approach is heavily inspired by EdgeBreaker, proposed by Rossignac in 1999. However, there is no mention or citation of this foundational paper. Although the authors provide a comparison in the supplementary material on Line 920, omitting a discussion of this algorithm in the main methods section does not adequately acknowledge its influence.
- Line 358: The experiments section begins directly with results, without any mention of training data, settings, or other critical experimental details. These details appear only in the supplementary material, which is insufficient, as they are essential for understanding the experiments.

**Questions:**

I have some questions primarily about the implementation and public availability of this work:

- Would it be possible to share the mesh tokenization script on an anonymous GitHub? This would help in understanding how the "traversal" process works and can be implemented.
- I didn’t find any mention in the paper regarding the public availability of the code and pre-trained model. Will these resources be released?

---

> ### Author Response · Authors · 2024-11-19
> **Response to Reviewer 8Uu4**
>
> Thank you for your valuable time and insightful comments! We have tried to address your concerns in the updated manuscript and our rebuttal text:
>
> **Q1: Presentation.**
>
> Thank you for the feedback! We have revised the manuscript to enhance its comprehensiveness:
>
> - Missing citations in the introduction have been added to appropriately acknowledge EdgeBreaker.
> - Due to space limitations in the main paper, we found it challenging to present both the original EdgeBreaker algorithm and our variant. Therefore, we have moved the detailed descriptions to the supplementary material and opted to use an example in the main paper to illustrate our method. Additionally, we have added text in the main paper to direct interested readers to the supplementary material for more details.
>
> **Q2: Public availability.**
>
> Thank you for your interest! We promise to release the code recently for better understanding and reproduction of our method.
>
> Unfortunately, the models may not be released at the same time due to dataset licensing issues of Objaverse-XL. However, we have made many test samples available on our project page, which should assist future methods in making comparisons.

---

> > ### Author Response · Authors · 2024-11-25
> > **Response to Reviewer 8Uu4**
> >
> > We sincerely appreciate your great efforts in reviewing this paper. Your constructive advice and valuable comments really help improve our paper. Considering the approaching deadline, please, let us know if you have follow-up concerns. We sincerely hope you can consider our reply in your assessment, and we can further address unclear explanations and remaining concerns if any.
> >
> > Once more, we are appreciated for the time and effort you've dedicated to our paper.

---

> > > ### Comment · Reviewer_8Uu4 · 2024-11-26
> > > **Response to authors**
> > >
> > > Thank authors for updating the manuscript and for the response.
> > >
> > > I noticed that the third point in the weaknesses I raised has been entirely ignored in the reply.
> > >
> > > Additionally, the explanation about dataset licensing issues remains unclear. Many other open-source projects do not seem to encounter problems when using Objaverse-XL. Could authors please clarify this matter in more detail?
> > > Furthermore, I am skeptical of the argument that the test samples provided on the project page can adequately support comparisons. Such an approach could potentially involve cherry-picked examples, leading to unfair comparisons. A more rigorous and transparent evaluation would be necessary to ensure fairness.
> > >
> > > Addressing these points thoroughly would not only strengthen the manuscript but also provide greater transparency and confidence in the results.

---

> ### Author Response · Authors · 2024-11-27
> **Response to Reviewer 8Uu4**
>
> Thanks for the reply!
>
> We apologize for the missing point in the presentation. To make it easier for readers to find information, we aim to keep all implementation details in a single location. However, due to page limits of the main paper, we are unable to include the full content without omitting critical information. Instead, we will include additional sentences in the final version to direct interested readers to the supplementary materials.
>
> Regarding the dataset issue, this paper is not solely an open-source project. To avoid anonymity violations and to the best of our knowledge, there are very few precedents of major corporations with formal legal departments releasing models trained on datasets with contentious licenses such as Objaverse-XL. Nevertheless, we are actively working to make the checkpoints available as soon as possible. Additionally, we have provided all necessary details, including final training losses, to ensure reproducibility.
>
> Lastly, we agree that a more extensive quantitative experiment is important. We conducted a new qualitative comparison experiment to evaluate the Chamfer Distance (CD) and Hausdorff Distance (HD) between the generated meshes and the input point clouds. Due to the lack of widely recognized evaluation datasets, we tried to construct two test datasets from different sources:
>
> - **AI-Mesh**: This dataset comprises 30 meshes generated by other image-to-3D models, extracted using the Marching Cubes algorithm.
> - **Artist-Mesh**: This dataset consists of 100 artistic meshes obtained from [basemesh](https://www.thebasemesh.com/), which were not seen during the training process.
>
> All meshes are normalized into $[-1, 1]^3$. The summarized evaluation scores are presented below:
>
> |             |                | MeshAnything | MeshAnythingV2 | Ours      |
> | ----------- | -------------- | ------------ | -------------- | --------- |
> | AI-Mesh     | CD$\downarrow$ | 0.072        | 0.081          | **0.034** |
> |             | HD$\downarrow$ | 0.170        | 0.241          | **0.079** |
> | Artist-Mesh | CD$\downarrow$ | 0.066        | 0.050          | **0.036** |
> |             | HD$\downarrow$ | 0.195        | 0.164          | **0.112** |
>
> We observed the following:
>
> - Our model achieves the best performance on surface metrics, even when using only point clouds as input. In contrast, the MeshAnything series further leverage surface normals.
> - MeshAnythingV2 performs significantly worse on the more challenging AI-Mesh dataset, often failing to generate complete surfaces in these cases (consistent with the results shown in Figure 5).
>
> As the deadline for editing the PDF is approaching, we will include these experiments in the final version of the paper.

---

### Official Review · Reviewer_7iBA · 2024-11-04

**Soundness:** 3
**Presentation:** 3
**Contribution:** 3
**Rating:** 6
**Confidence:** 4

**Summary:**

This paper focuses on generating artistic meshes with up to 4000 faces and a resolution of 512³. It introduces a novel mesh tokenization algorithm based on EdgeBreaker to facilitate lossless face compression. To produce fixed-length latent codes for arbitrary mesh sequences, the authors propose an encoder alongside an auto-regressive decoder. Additionally, they investigate the incorporation of image conditions to guide the generation of the meshes.

**Strengths:**

1. The proposed method can generate meshes with up to 4000 faces, surpassing the capabilities of previous baselines.
2. The trained model shows strong generalization on novel inputs.
3. Extensive experiments highlight the advantages of the proposed method in achieving high-quality mesh generation.

**Weaknesses:**

1. The proposed method and the baselines are trained on different datasets (for example, MeshAnything does not have access to Objaverse-XL and reserves 10% of Objaverse for evaluation). As a result, the comparisons can be unfair.
2. Is the training sequence unique for each mesh? How did you define the start of the sequence?
3. Although the authors report the inference speed, there is no comparison of the inference speed against other methods when generating similar number of faces.
4. The details of user study are missing. How many users are there in the test? What is the details of the task or questions that volunteers are asked to do or answer? Are 8 test cases randomly selected? It is better to have more test cases rather than only 8 ones.
5. To demonstrate the robustness across all test cases, it is better to involve quantitative evaluations like those in *MeshAnything*, where the authors sample points from both the generated mesh and the ground-truth mesh, calculating the CD or ECD.

**Questions:**

1. What does the dashed line mean in Figure 2?
2. Does the output number of faces strictly adhere to the face count control? For instance, when using a control token of 1000 to 2000, will the model generate only mesh sequences within that range?

---

> ### Author Response · Authors · 2024-11-19
> **Response to Reviewer 7iBA**
>
> Thank you for your valuable time and insightful comments! We have tried to address your concerns in the updated manuscript and our rebuttal text:
>
> **Q1: Unfair comparisons due to different training datasets.**
>
> We acknowledge the difficulty in ensuring that the datasets used are identical. MeshAnything does not release the specific subset list of Objaverse and additionally incorporates data from ShapeNet. Moreover, our model's capability to support longer sequences enables us to utilize a greater number of samples from Objaverse, which plays a critical role in enhancing performance.
>
> **Q2: Is the training sequence unique for each mesh? How did you define the start of the sequence?**
>
> Yes, similar to previous works, we sort the faces in an empirical $y\text{-}z\text{-}x$ order and always begin with the first half-edge of the first face. The EdgeBreaker algorithm is deterministic when the starting half-edge is fixed, which ensures that the sequence is unique for each mesh.
>
> **Q3: Comparison of inference speed.**
>
> Thanks for the advice! Since MeshAnything and MeshAnythingV2 do not support face count control, we report only the average generation speed:
>
> |             | MeshAnything | MeshAnythingV2 | Ours   |
> | ----------- | ------------ | -------------- | ------ |
> | Tokens/s    | 108.53       | 112.55         | 103.49 |
> | Triangles/s | 12.06        | 28.03          | 24.44  |
>
> Due to the larger number of parameters in our model, the generation speed is slightly slower compared to MeshAnythingV2. However, our model demonstrates greater robustness and delivers better performance.
>
> **Q4: Details on user study.**
>
> We have revised the manuscript to include additional details about the user study. Volunteers were asked to evaluate the results based on three criteria: geometry consistency with the input point cloud, visual appearance of the triangle faces, and overall mesh quality. Each volunteer was presented with mixed results from randomly selected methods. Additionally, we have provided more samples—68 point-cloud-conditioned and 52 image-conditioned meshes—generated by our method on our project page for better comparisons.
>
> **Q5: Lack of quantitative evaluations like CD or ECD.**
>
> Thanks for the advice! We believe that surface metrics may not effectively capture the aesthetic quality of the mesh, which is the primary objective of auto-regressive mesh generation. Additionally, on our more challenging test dataset, baseline methods often fail noticeably and are unable to produce complete mesh surfaces, further diminishing the relevance of surface metrics in evaluating performance.
>
> **Q6: Meaning of the dashed line in Figure 2.**
>
> The dashed line indicates that our image-conditioned diffusion model is trained separately. Initially, we train the point-conditioned model (ArAE), as depicted above the dashed line, and subsequently use the learned fixed-length latent space to train the diffusion model, as shown below the dashed line.
>
> **Q7: Adherence of face count control.**
>
> Since our face count control is learned implicitly, the model cannot be strictly constrained to adhere to the specified range. We employ four learnable tokens to control the face count: unconditional, $(0, 1000)$, $[1000, 2000)$, $[2000, 4000)$. Our observations indicate that the control token $[1000, 2000)$ provides the most robust face count control and generation. In contrast, the other tokens exhibit less robustness and may also be influenced by the complexity of the input shape.

---

> > ### Author Response · Authors · 2024-11-25
> > **Response to Reviewer 7iBA**
> >
> > We sincerely appreciate your great efforts in reviewing this paper. Your constructive advice and valuable comments really help improve our paper. Considering the approaching deadline, please, let us know if you have follow-up concerns. We sincerely hope you can consider our reply in your assessment, and we can further address unclear explanations and remaining concerns if any.
> >
> > Once more, we are appreciated for the time and effort you've dedicated to our paper.

---

> > > ### Comment · Reviewer_7iBA · 2024-11-26
> > >
> > > The authors have addressed most of my concerns and questions. However, I strongly recommend including quantitative evaluations, such as CD (or other suitable metrics), on a larger number of samples. This would help demonstrate robustness across a broader range of cases and rule out the possibility of cherry-picked results. Besides, I suggest including the comparison of inference speed in the manuscript for the final version to help the readers have a better understanding.

---

> ### Author Response · Authors · 2024-11-27
> **Response to Reviewer 7iBA**
>
> Thanks for the reply!
>
> Following your advice, we conducted a new qualitative comparison experiment to evaluate the Chamfer Distance (CD) and Hausdorff Distance (HD) between the generated meshes and the input point clouds. Due to the lack of widely recognized evaluation datasets, we tried to construct two test datasets from different sources:
>
> - **AI-Mesh**: This dataset comprises 30 meshes generated by other image-to-3D models, extracted using the Marching Cubes algorithm.
> - **Artist-Mesh**: This dataset consists of 100 artistic meshes obtained from [basemesh](https://www.thebasemesh.com/), which were not seen during the training process.
>
> All meshes are normalized into $[-1, 1]^3$. The summarized evaluation scores are presented below:
>
> |             |                | MeshAnything | MeshAnythingV2 | Ours      |
> | ----------- | -------------- | ------------ | -------------- | --------- |
> | AI-Mesh     | CD$\downarrow$ | 0.072        | 0.081          | **0.034** |
> |             | HD$\downarrow$ | 0.170        | 0.241          | **0.079** |
> | Artist-Mesh | CD$\downarrow$ | 0.066        | 0.050          | **0.036** |
> |             | HD$\downarrow$ | 0.195        | 0.164          | **0.112** |
>
> We observed the following:
>
> - Our model achieves the best performance on surface metrics, even when using only point clouds as input. In contrast, the MeshAnything series further leverage surface normals.
> - MeshAnythingV2 performs significantly worse on the more challenging AI-Mesh dataset, often failing to generate complete surfaces in these cases (consistent with the results shown in Figure 5).
>
> As the deadline for editing the PDF is approaching, we will include these experiments in the final version of the paper.

---

### Official Review · Reviewer_kGAF · 2024-11-04

**Soundness:** 2
**Presentation:** 3
**Contribution:** 2
**Rating:** 3
**Confidence:** 4

**Summary:**

The paper introduces EdgeRunner, an Auto-regressive Auto-encoder (ArAE) model for artistic mesh generation. It proposes a mesh tokenization algorithm to address incompleteness and poor generalization.

The model can generate 3D meshes with up to 4,000 faces at a spatial resolution 512^3. It also compresses variable-length meshes into a fixed-length latent space, enabling the training of latent diffusion models for better generalization.

The main contributions include a mesh tokenization algorithm for lossless face compression, an ArAE model for fixed-length latent code generation, the use of this latent space for training latent diffusion models with better generalization.

**Strengths:**

EdgeRunner demonstrates strengths in the mesh tokenization algorithm and the Auto-regressive Auto-encoder framework, which address previous challenges in mesh generation.

It offers a mesh tokenization algorithm for efficient compression into 1D token sequences, enabling mesh generation with up to 4,000 faces at a 512^3 resolution. The model's ability to compress variable-length meshes into a fixed-length latent space facilitates the training of latent diffusion models for enhanced generalization.

**Weaknesses:**

A. While the paper presents a compelling case for EdgeRunner's capabilities, a notable weakness is that it does not fundamentally differentiate itself from existing works like MeshGPT and the MeshAnything series in terms of the core generative approach.

B. Despite the improvements in tokenization and the auto-regressive framework, the generated 3D geometries may not adhere to the modeling and wiring conventions that human artists typically follow. This could potentially limit the acceptance and practical application of the generated meshes within professional 3D content creation pipelines, where adherence to standard modeling practices is often a requirement.

C. The paper also has a limitation in terms of polycount, as the capability of generating meshes with up to 4,000 faces, while an improvement over previous methods, still falls short for many real-world scenarios. The constraint on the number of faces limits the model's ability to capture the complexity and detail required for high-fidelity 3D representations, which further limits the application.

D. Another limitation of the paper is the potentially unfair comparison made with Unique3D, as the latter employs a multi-view approach to 3D generation. Unique3D's methodology is inherently challenged by inconsistencies or deformities that can arise from generating a 3D model from multiview images, which is a different set of problems compared to EdgeRunner. This discrepancy means that the comparison may not accurately reflect the strengths and weaknesses of each method, as they are optimized for different conditions and face distinct sets of challenges. Consequently, the paper's claims about EdgeRunner's superiority might not be fully substantiated.

**Questions:**

I have a few questions for the authors to consider:

The paper presents an improvement in the number of faces that can be generated compared to previous methods, reaching up to 4,000 faces. However, for many practical applications in industries such as gaming, film, etc, this number is still limited. How to increase the face count further? Are there any technical barriers within the current model that restrict the generation of meshes with even more faces?

Comparison with Unique3D is not fair. How does the method perform when compared with CLAY? How does EdgeRunner compare in terms of quality and diversity of generated meshes? While the code CLAY is not publicly available, its geometric rendering effects can be observed on Hugging Face and its corresponding products are available for comparison.

---

> ### Author Response · Authors · 2024-11-19
> **Response to Reviewer kGAF**
>
> Thank you for your valuable time and insightful comments! We have tried to address your concerns in the updated manuscript and our rebuttal text:
>
> **Q1: Difference from MeshGPT and MeshAnything in terms of the core generative approach.**
>
> Our core contribution lies in a novel mesh tokenizer designed for improved compression of mesh sequences, which enhances training efficiency and boosts performance. For the core generative approach, auto-regressive mesh generation methods all rely on transformers and next-token prediction.
>
> **Q2: The generated geometry may not adhere to the conventions that artists follow.**
>
> The goal of auto-regressive mesh generation is to learn and replicate artistic mesh conventions from large datasets. We believe our work advances this objective and can generate meshes that are indistinguishable from artist-created ones.
>
> **Q3: Limited number of generated faces.**
>
> As discussed in the paper, although 4000 faces may not suffice for complex objects, this count is adequate for many common objects, as demonstrated in both the paper and on the project page. Moreover, previous methods are limited to generating up to 1600 faces with lower robustness.
>
> To further increase the face count, one approach is to develop improved tokenization algorithms that achieve a higher compression rate. Another possible direction is to employ local attention and perform sliding-window inference, based on the observation that triangle topology relies more on local features than on global features.
>
> **Q4: Unfair comparison with Unique3D.**
>
> We argue that our work is the first to achieve end-to-end image-conditioned artistic mesh generation. Existing image-to-3D methods continue to depend on isosurfacing techniques, such as Marching Cubes, to extract dense meshes, which fundamentally differ from our approach. We selected Unique3D for comparison as it is among the most recent open-source works in the field of image-to-3D, and this comparison aims to highlight the differences between dense meshes and artistic meshes.
>
> In this sense, CLAY also produces dense meshes extracted using Marching Cubes similar to Unique3D, which are far from artistic meshes. We acknowledge that our approach may offer less geometric detail and generalization capability compared to 3D diffusion-based methods; however, this is not our primary focus. As in your own words, we believe "the comparison may not accurately reflect the strengths and weaknesses of each method, as they are optimized for different conditions and face distinct sets of challenges".

---

> > ### Author Response · Authors · 2024-11-25
> > **Response to Reviewer kGAF**
> >
> > We sincerely appreciate your great efforts in reviewing this paper. Your constructive advice and valuable comments really help improve our paper. Considering the approaching deadline, please, let us know if you have follow-up concerns. We sincerely hope you can consider our reply in your assessment, and we can further address unclear explanations and remaining concerns if any.
> >
> > Once more, we are appreciated for the time and effort you've dedicated to our paper.

---

### Meta-Review · Area_Chair_LwwF · 2024-12-17

**Metareview:**

The paper introduces EdgeRunner, an auto-regressive auto-encoder with a novel mesh tokenization algorithm for high-quality artistic mesh generation, demonstrating significant advancements in mesh compression and fixed-length latent code learning. Reviewers praised the paper's technical contributions, including the innovative use of EdgeBreaker-inspired tokenization, strong experimental results, and the ability to generalize to point clouds and single-view image inputs, producing visually appealing meshes. However, concerns were raised about limited quantitative evaluations, lack of comparisons to alternative methods like CLAY, the relatively low face count (4,000 faces), and the need for clearer explanations of training settings and dataset usage in the main text.

**Additional Comments On Reviewer Discussion:**

During the rebuttal period, reviewers raised concerns about the lack of quantitative evaluations (e.g., Chamfer Distance), fairness of comparisons with alternative methods, insufficient experimental details, and limited face count for real-world applications. The authors addressed these by conducting new experiments to evaluate Chamfer and Hausdorff Distances on two test datasets, clarifying dataset usage, inference speed, and robustness, while also revising the manuscript to include more details on user studies and training processes. Despite some lingering concerns about dataset licensing and comparisons, reviewers acknowledged the improvements and found most issues satisfactorily resolved.

---

### Decision · Program_Chairs · 2025-01-22

Accept (Poster)